# HYPER-SAGNN: A SELF-ATTENTION BASED GRAPH NEURAL NETWORK FOR HYPERGRAPHS

**Ruochi Zhang**
School of Computer Science
Carnegie Mellon University

**Yuesong Zou**
School of Computer Science
Carnegie Mellon University
IIIS, Tsinghua University

**Jian Ma**
School of Computer Science
Carnegie Mellon University
jianma@cs.cmu.edu

## ABSTRACT

Graph representation learning for hypergraphs can be used to extract patterns among higher-order interactions that are critically important in many real world problems. Current approaches designed for hypergraphs, however, are unable to handle different types of hypergraphs and are typically not generic for various learning tasks. Indeed, models that can predict variable-sized heterogeneous hyperedges have not been available. Here we develop a new self-attention based graph neural network called Hyper-SAGNN applicable to homogeneous and heterogeneous hypergraphs with variable hyperedge sizes. We perform extensive evaluations on multiple datasets, including four benchmark network datasets and two single-cell Hi-C datasets in genomics. We demonstrate that Hyper-SAGNN significantly outperforms the state-of-the-art methods on traditional tasks while also achieving great performance on a new task called outsider identification. Hyper-SAGNN will be useful for graph representation learning to uncover complex higher-order interactions in different applications.

## 1 INTRODUCTION

Graph structure is a widely used representation for data with complex interactions. Learning on graphs has also been an active research area in machine learning on how to predict or discover patterns based on the graph structure (Hamilton et al., 2017b). Although existing methods can achieve strong performance in tasks such as link prediction and node classification, they are mostly designed for analyzing pair-wise interactions and thus are unable to effectively capture higher-order interactions in graphs. In many real-world applications, however, relationships among multiple instances are key to capturing critical properties, e.g., co-authorship involving more than two authors or relationships among multiple heterogeneous objects such as "(human, location, activity)". Hypergraphs can be used to represent higher-order interactions (Zhou et al., 2007). To analyze higher-order interaction data, it is straightforward to expand each hyperedge into pair-wise edges with the assumption that the hyperedge is decomposable.

Several previous methods were developed based on this notion (Sun et al., 2008; Feng et al., 2018). However, earlier work DHNE (Deep Hyper-Network Embedding) (Tu et al., 2018) suggested the existence of heterogeneous indecomposable hyperedges where relationships within an incomplete subset of a hyperedge do not exist. Although DHNE provides a potential solution by modeling the hyperedge directly without decomposing it, due to the neural network structure used in DHNE, the method is limited to the fixed type and fixed-size heterogeneous hyperedges and is unable to consider relationships among multiple types of instances with variable size. For example, Fig. 1 shows a heterogeneous co-authorship hypergraph with two types of nodes (corresponding author and coauthor). Due to the variable number of both authors and corresponding authors in a publication, the hyperedges (co-authorship) have different sizes or types. Unfortunately, methods for representation learning of heterogeneous hypergraph with variable-sized hyperedges, especially those that can predict variable-sized hyperedges, have not been developed.

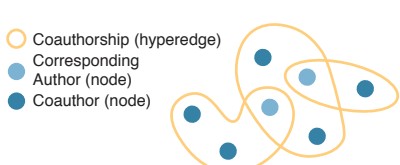

**Figure 1:** An example of the co-authorship hypergraph. Here authors are represented as nodes (in dark blue and light blue) and coauthorships are represented as hyperedges.

In this work, we developed a self-attention based graph neural network, called Hyper-SAGNN that can work with both homogeneous and heterogeneous hypergraphs with variable hyperedge size. Using the same datasets in the DHNE paper (Tu et al., 2018), we demonstrated the advantage of Hyper-SAGNN over DHNE in multiple tasks. We further tested the effectiveness of the method in predicting edges and hyperedges and showed that the model can achieve better performance from the multi-tasking setting. We also formulated a novel task called outsider identification and showed that Hyper-SAGNN performs strongly. Importantly, as an application of Hyper-SAGNN to single-cell genomics, we were able to learn the embeddings for the most recently produced single-cell Hi-C (scHi-C) datasets to uncover the clustering of cells based on their 3D genome structure (Ramani et al., 2017; Nagano et al., 2017). We showed that Hyper-SAGNN achieved improved results in identifying distinct cell populations as compared to existing scHi-C clustering methods. Taken together, Hyper-SAGNN can significantly outperform state-of-the-art methods and can be applied to a wide range of hypergraphs for different applications.

## 2 RELATED WORK

Deep learning based models have been developed recently to generalize from graphs to hypergraphs (Gui et al., 2016; Tu et al., 2018). The HyperEdge Based Embedding (HEBE) method (Gui et al., 2016) aims to learn the embeddings for each object in a specific heterogeneous event by representing it as a hyperedge. However, as demonstrated in Tu et al. (2018), HEBE does not perform well on sparse hypergraphs. Notably, previous methods typically decompose the hyperedge into pair-wise relationships where the decomposition methods can be divided into two categories: explicit and implicit. For instance, given a hyperedge $(v_1, v_2, v_3)$, the explicit approach would decompose it directly into three edges, $(v_1, v_2), (v_2, v_3), (v_1, v_3)$, while the implicit approach would add a hidden node $e$ representing the hyperedge before decomposition, i.e., $(v_1, e), (v_2, e), (v_3, e)$. The deep hypergraph embedding (DHNE) model, however, directly models the tuple-wise relationship using MLP (Multilayer Perceptron). The method is able to achieve better performance on multiple tasks as compared to other methods designed for graphs or hypergraphs such as Deepwalk (Perozzi et al., 2014), node2vec (Grover & Leskovec, 2016), and HEBE. Unfortunately, the structure of MLP takes fixed-size input, making DHNE only capable of handling $k$-uniform hypergraphs, i.e., hyperedges containing $k$ nodes. To use DHNE for non-$k$-uniform hypergraphs or hypergraphs with different types of hyperedges, a function for each type of hyperedges needs to be trained individually, which leads to significant computational cost and loss of the capability to generalize to unseen types of hyperedges. Similarly, heterogeneous hyper-network embedding (Baytas et al., 2018) also used MLP as part of the model which requires fixed size input to train the model. Another recent method, hyper2vec (Huang et al., 2019), can also generate embeddings for nodes within the hypergraph and outperforms other hypergraph embedding methods such as HGE (Yu et al., 2018) in the node classification task. However, hyper2vec cannot solve the link prediction problem directly as it only generates the embeddings of nodes in an unsupervised manner without a learned function to map from embeddings of nodes to hyperedges. Also, for $k$-uniform hypergraphs, hyper2vec is equivalent to node2vec, which cannot capture the high-order network structures for indecomposable hyperedges (as shown in Tu et al. (2018)). Moreover, graph neural network based methods (Yadati et al., 2018; Feng et al., 2019; Bai et al., 2019) have been proposed to generalize the convolution operation or attention mechanism from graphs to hypergraphs. However, these methods mainly focus on the hypergraphs where node attributes are known and are typically used for semi-supervised node classification tasks. Similar to hyper2vec, these methods cannot be directly used for predicting hyperedges. Our Hyper-SAGNN in this work addresses all these challenges with a self-attention based graph neural network that can learn embeddings of the nodes and predict hyperedges for non-$k$-uniform heterogeneous hypergraphs.

## 3 METHOD

### 3.1 DEFINITIONS AND NOTATIONS

**Definition 1. (Hypergraph)** A hypergraph is defined as $G = (V, E)$, where $V = \{v_1, ..., v_n\}$ represents the set of nodes in the graph, and $E = \{e_i = (v_1^{(i)}, ..., v_k^{(i)})\}$ represents the set of hyperedges. For any hyperedge $e$, it can contain more than two nodes (i.e., $\delta(e) \geq 2$). If all hyperedges within a hypergraph have the same size of $k$, it is called a $k$-uniform hypergraph. Note

that even if a hypergraph is $k$-uniform, it can still have different types of hyperedges because the node type can vary for nodes within the hyperedges.

**Definition 2. (The hyperedge prediction problem)** We formally define the hyperedge prediction problem. For a given tuple $(v_1, v_2, ..., v_k)$, our goal is to learn a function $f$ that satisfies:

$$f(v_1, v_2, ..., v_k) = \begin{cases} \geq s, & \text{if } (v_1, v_2, ..., v_k) \in E \\ < s, & \text{if } (v_1, v_2, ..., v_k) \notin E \end{cases} \tag{1}$$

where $s$ is the threshold to binarize the continuous value of $f$ into a label, which indicates whether the tuple is an hyperedge or not. Specifically, when we are given the pre-trained embedding vectors or the features of nodes $X = \{x_1, ..., x_i\}$, we can rewrite this function as:

$$f(v_1, v_2, ..., v_k) \triangleq f(g(x_1), g(x_2), ..., g(x_k)) \tag{2}$$

where the vectors $g(x_i)$ can be considered as the fine-tuned embedding or embedding vectors for the nodes. For convenience, we refer to $x_i$ as the features and $g(x_i)$ as the learned embeddings.

## 3.2 STRUCTURE OF HYPER-SAGNN

Our goal is to learn the functions $f$ and $g$ that take tuples of node features $(x_1, ..., x_k)$ as input and produce the probability of these nodes forming a hyperedge. Without the assumption that the hypergraph is $k$-uniform and the type of each hyperedge is identical, we require that $f$ can take variable-sized, non-ordered input. Although simple functions such as average pooling $f(g(x_1), ..., g(x_k)) = \frac{1}{K} \sum_{i=1}^{k} g(x_i)$ satisfy this tuple-wise condition, previous work showed that the linear function is not sufficient to model this relationship (Tu et al., 2018). DHNE used an MLP to model the non-linear function, but it requires that an individual function needs to be trained for different types of hyperedges. Here we propose a new method to tackle the general hyperedge prediction problem.

Graph neural network based methods such as GraphSAGE (Hamilton et al., 2017a) typically define a unique computational graph for each node, allowing it to perform efficient information aggregation for nodes with different degrees. Graph Attention Network (GAT) (Veličković et al., 2017) utilizes a self-attention mechanism in the information aggregation process. Motivated by these properties, we propose our method Hyper-SAGNN based on the self-attention mechanism within each tuple to learn the function $f$.

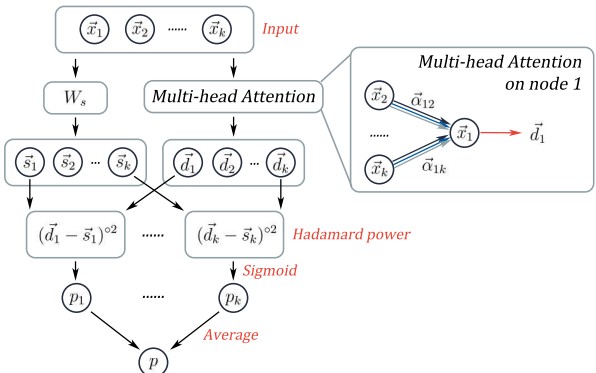

**Figure 2:** Structure of the neural network used in Hyper-SAGNN. The input $(\vec{x}_1, \vec{x}_2, ..., \vec{x}_k)$, representing the features for nodes 1 to $k$, passes through two branches of the network resulting in static embeddings $(\vec{s}_1, \vec{s}_2, ..., \vec{s}_k)$ and dynamic embeddings $(\vec{d}_1, \vec{d}_2, ..., \vec{d}_k)$, respectively. The layer for generating dynamic embeddings is the multi-head attention layer. An example for its mechanism on node 1 here is shown in the figure as well. Then the pseudo-euclidean distance of each pair of static and dynamic embeddings is calculated by one-layered position-wise feed-forward network to produce probability scores $(p_1, p_2, ..., p_k)$. These scores are further averaged to represent whether this group of nodes form a hyperedge.

We first briefly introduce the self-attention mechanism. We use the same terms as the self-attention mechanism described in Vaswani et al. (2017); Veličković et al. (2017). Given a group of nodes $(\vec{x}_1, \vec{x}_2, ..., \vec{x}_k)$ and weight matrices $W_Q, W_K, W_V$ that represent linear transformations of features before applying the dot-product attention to be trained, we first compute the attention coefficients that reflect the pairwise importance of nodes:

$$e_{ij} = \left(W_Q^T x_i\right)^T \left(W_K^T x_j\right), \forall 1 \leq i, j \leq k \tag{3}$$

We then normalize $e_{ij}$ by all possible $j$ within the tuple through the softmax function, i.e.,

$$\alpha_{ij} = \frac{\exp(e_{ij})}{\sum_{1 \leq l \leq k} \exp(e_{il})} \tag{4}$$

Finally, a weighted sum of the transformed features with an activation function is calculated:

$$\vec{d_i} = \tanh\left(\sum_{1 \leq j \leq k, i \neq j} \alpha_{ij} W_V^T x_j\right) \tag{5}$$

In GAT, each node is applied to the self-attention mechanism usually with all its first-order neighbors. In Hyper-SAGNN, we aggregate the information for a node $v_i$ only with its neighbors for a given tuple. The structure of Hyper-SAGNN is illustrated in Fig. 2.

The input to our model can be represented as tuples, i.e., $(\vec{x}_1, \vec{x}_2, ..., \vec{x}_k)$. Each tuple first passes through a position-wise feed-forward network to produce $(\vec{s}_1, \vec{s}_2, ..., \vec{s}_k)$, where $\vec{s}_i = \tanh(W_s^T \vec{x}_i)$. We refer to each $\vec{s}_i$ as the static embedding for node $i$ since it remains the same for node $i$ no matter what the given tuple is. The tuple also passes through a multi-head graph attention layer to produce a new set of node embedding vectors $(\vec{d}_1, \vec{d}_2, ..., \vec{d}_k)$, which we refer to as the dynamic embeddings because they are dependent on all the node features within this tuple.

Note that unlike the standard attention mechanism described above, when calculating $\vec{d}_i$, we require that $j \neq i$ in Eqn. (5). In other words, we exclude the term $\alpha_{ii} W_V^T x_i$ in the calculation of dynamic embeddings. Based on our results we found that including $\vec{\alpha}_{ii}$ would lead to either similar or worse performance in terms of hyperedge prediction and node classification (see Appendix A.6 for details). We will elaborate on the motivation of this choice later in this section.

With the static and dynamic embedding vectors for each node, we calculate the Hadamard power (element-wise power) of the difference of the corresponding static/dynamic pair. It is then further passed through a one-layered neural network with sigmoid as the activation function to produce a probability score $p_i$. Finally, all the output $p_i \in [0, 1]$ is averaged to get the final $p$, i.e.,

$$o_i = W_o^T((\vec{d}_i - \vec{s}_i)^{\circ 2}) + b \tag{6}$$

$$p = \frac{1}{K}\sum_{i=1}^{k} p_i = \frac{1}{K}\sum_{i=1}^{k} \sigma(o_i) \tag{7}$$

By design, $o_i$ can be regarded as the squared weighted pseudo-euclidean distance between the static embedding $\vec{s}_i$ and the dynamic one $\vec{d}_i$. It is called pseudo-euclidean distance because we do not require the weight to be non-zero or to sum up to 1. One rationale for allowing negative weights when calculating the distance could be the Minkowski space where the distance is defined as $d^2 = x^2 + y^2 + z^2 - t^2$. Therefore, for these high-dimensional embedding vectors, we do not specifically treat them as euclidean vectors.

Our network essentially aims to build the correlation of the average "distance" of the static/dynamic embedding pairs with the probability of the node group forming a hyperedge. Since the dynamic embedding is the weighted sum of features (with potential non-linear transformation) from neighbors within the tuple, this "distance" reflects how well the static embedding of each node can be approximated by the features of their neighbors within that tuple. This design strategy shares some similarities with the CBOW model in natural language processing (Mikolov

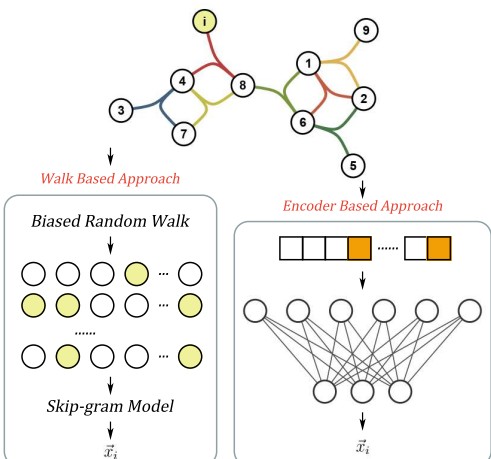

**Figure 3:** Illustration of the method for generating node features for node $i$ in the hypergraph. In the walk based approach, a biased random walk on hypergraphs is used to produce walking paths (the yellow circles in the walking paths represent node $i$). These walks are further used to train a skip-gram model to generate features. In the encoder based approach, the $i$-th row of the adjacency matrix (as shown in the figure where the orange/white blocks represent whether or not node $i$ is adjacent to other nodes in the graph) is used as the input to an autoencoder. The output of the encoder part is used as the features for node $i$.

et al., 2013), where the model aims to predict the target word given its context (see Appendix A.5 for the analysis of the static/dynamic embedding pairs). In principle, we could still include the $\vec{\alpha}_{ii}$

term to obtain the embedding $\vec{d}_i^*$. Alternatively, we can directly pass $\vec{d}_i^*$ through a fully connected layer to produce $p_i^*$ while the rest remains the same. However, we argue that our proposed model would be able to produce $s_i$ that can be directly used for tasks such as node classification while the alternative approach is unable to achieve that (see Appendix A.6 for detailed analysis).

### 3.3 Approaches for generating features

In an inductive learning setting with known attributes for the nodes, $\vec{x}_i$ can just be the attributes of the node. However, in a transductive learning setting without knowing the attributes of the nodes, we have to generate $\vec{x}_i$ based on the graph structure solely. Here we use two existing strategies to generate features $\vec{x}_i$. As shown in Fig. 3, the first approach is the random walk based method. We designed a biased random walk scheme for nodes in hypergraphs and used that to sample walks. Then, similar to node2vec, a skip-gram model is trained to generate features. The second approach is the encoder based approach where the corresponding row of the adjacency matrix is used as the features. The features are further passed through an autoencoder-like structure to reduce the dimensionality with the output of the hidden layer used as the features. The detailed description of these two approaches can be found in Appendix A.1.

## 4 Results

We sought to compare Hyper-SAGNN with the state-of-the-art method DHNE as it has already been demonstrated with superior performance over previous algorithms such as DeepWalk, LINE, and HEBE. We first used the same four datasets in the DHNE paper to have a direct comparison. The details on these datasets can be found in the Appendix A.2. The details on the parameters used in this section for both Hyper-SAGNN and other methods can also be found in the Appendix A.3. The details of the tasks and the evaluation metrics used in this section are explained in the Appendix A.4.

### 4.1 Performance comparison with existing methods

We evaluated the effectiveness of our embedding vectors and the learned function with the network reconstruction task. We compared our Hyper-SAGNN using the encoder based approach and also the model using the random walk based pre-trained embeddings against DHNE and the baseline node2vec. We first trained the model and then used the learned embeddings to predict the hyperedges of the original network. We sampled the negative samples to be 5 times the amount of the positive samples following the same setup of DHNE. We evaluated the performance based on both the AUROC (Area Under the Receiver Operating Characteristic curve) and the AUPR (Area under the Precision-Recall curve). As shown in Table 1, Hyper-SAGNN can capture the network structure better than DHNE over all datasets either using the encoder based approach or the random walk based approach.

**Table 1:** AUC and AUPR values for network reconstruction. The models trained with the random walk based approach and the encoder based approach are marked as Hyper-SAGNN-W and Hyper-SAGNN-E, respectively.

|  | GPS | | MOVIELENS | | DRUG | | WORDNET | |
| --- | --- | --- | --- | --- | --- | --- | --- | --- |
|  | AUC | AUPR | AUC | AUPR | AUC | AUPR | AUC | AUPR |
| node2vec-mean | 0.572 | 0.188 | 0.557 | 0.197 | 0.668 | 0.246 | 0.613 | 0.215 |
| node2vec-min | 0.570 | 0.187 | 0.535 | 0.186 | 0.682 | 0.257 | 0.576 | 0.201 |
| DHNE | 0.959 | 0.836 | 0.974 | 0.878 | 0.952 | 0.873 | 0.989 | 0.953 |
| Hyper-SAGNN-E | 0.971 | **0.877** | 0.991 | 0.952 | 0.977 | 0.916 | 0.989 | 0.950 |
| Hyper-SAGNN-W | **0.976** | 0.857 | **0.998** | **0.986** | **0.988** | **0.945** | **0.994** | **0.956** |

We further assessed the performance of Hyper-SAGNN on the hyperedge prediction task. We randomly split the hyperedge set into training and testing set by a ratio of 4:1. The way to generate negative samples is the same as the network reconstruction task. As shown in Table 2, our model again achieves significant improvement over DHNE for predicting the unseen hyperedges. The most significant improvement is from the wordnet dataset, which is about a 24.6% increase on the AUPR score. For network reconstruction and hyperedge prediction tasks, the difference between the random walk based Hyper-SAGNN and the encoder based Hyper-SAGNN is minor.

In addition to the tasks related to the prediction of hyperedges, we also evaluated whether the learned embeddings are effective for node classification tasks. A multi-label classification experiment and a multi-class classification experiment were carried out for the MovieLens dataset and the wordnet dataset, respectively. We used Logistic Regression as the classifier. The proportion of the training

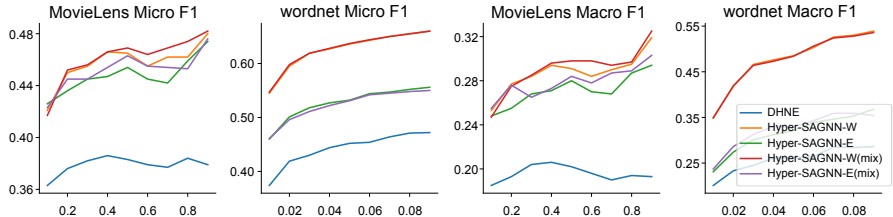

**Figure 4:** Performance of classification on MovieLens and wordnet datasets. Hyper-SAGNNs trained with the random walk based approach and the encoder based approach are marked as Hyper-SAGNN-W, Hyper-SAGNN-E, respectively. The models trained with a mix of edges and hyperedges are denoted with "(mix)".

data was chosen to be from 10% to 90% for the MovieLens dataset, and 1% to 10% for the wordnet dataset. We used averaged Mirco-F1 and Macro-F1 to evaluate the performance. The results are in Fig. 4. We observed that Hyper-SAGNN consistently achieves both higher Micro-F1 and Macro-F1 scores over DHNE for different fractions of the training data. Also, Hyper-SAGNN based on the random walk generally achieves the best performance (Hyper-SAGNN-W in Fig. 4).

## 4.2 PERFORMANCE ON NON-$k$-UNIFORM HYPERGRAPH

Next, we evaluated Hyper-SAGNN using the non-$k$-uniform heterogeneous hypergraph. For the above four datasets, we decomposed each hyperedge into 3 pairwise edges and added them to the existing graph. We trained our model to predict both the hyperedges and the edges (i.e., non-hyperedges). We then evaluated the performance for link prediction tasks for both the hyperedges and the edges. We also performed the node classification task following the same setting as above. The results for link prediction are in Table 2. Fig. 4 shows the results for the node classification task.

**Table 2:** Performance evaluation based on AUROC and AUPR for hyperedge/edge prediction. Methods with annotation (mix) represent Hyper-SAGNN trained with a mixture of edges and hyper-edges. Datasets marked with "(2)" represent the performance on pair-wise edge prediction (i.e., non-hyperedges).

|  | GPS | | MOVIELENS | | DRUG | | WORDNET | |
| --- | --- | --- | --- | --- | --- | --- | --- | --- |
|  | AUC | AUPR | AUC | AUPR | AUC | AUPR | AUC | AUPR |
| node2vec - mean | 0.563 | 0.191 | 0.562 | 0.197 | 0.670 | 0.246 | 0.608 | 0.213 |
| node2vec - min | 0.570 | 0.185 | 0.539 | 0.186 | 0.684 | 0.258 | 0.575 | 0.200 |
| DHNE | 0.910 | 0.668 | 0.877 | 0.668 | 0.925 | 0.859 | 0.816 | 0.459 |
| Hyper-SAGNN-E | **0.952** | **0.798** | 0.926 | 0.793 | **0.961** | **0.895** | **0.890** | 0.705 |
| Hyper-SAGNN-W | 0.922 | 0.722 | **0.930** | **0.810** | 0.955 | 0.892 | 0.880 | **0.706** |
| Hyper-SAGNN-E (mix) | 0.950 | 0.795 | 0.928 | 0.799 | 0.956 | 0.887 | 0.881 | 0.694 |
| Hyper-SAGNN-W (mix) | 0.920 | 0.720 | 0.929 | 0.811 | 0.950 | 0.889 | 0.884 | 0.684 |
|  | GPS (2) | | MOVIELENS (2) | | DRUG (2) | | WORDNET (2) | |
|  | AUC | AUPR | AUC | AUPR | AUC | AUPR | AUC | AUPR |
| Hyper-SAGNN-E (mix) | 0.921 | 0.899 | 0.971 | 0.967 | 0.981 | 0.973 | 0.891 | 0.897 |
| Hyper-SAGNN-W (mix) | 0.931 | 0.910 | 0.999 | 0.999 | 0.999 | 0.999 | 0.923 | 0.916 |

We observed that Hyper-SAGNN can preserve the graph structure on different levels. Compared to training the model with hyperedges only, including the edges into the training would not cause obvious changes in performance for hyperedge predictions (about a 1% fluctuation for AUC/AUPR).

We then further assessed the model in a new evaluation setting where there are adequate edges but only a few hyperedges presented. We asked whether the model can still achieve good performance on the hyperedge prediction based on this dataset. This scenario is possible in real-world applications especially when the dataset is combined from different sources. For example, in the drug dataset, it is possible that, in addition to the (user, drug, reaction) hyperedges, there are also extra edges that come from other sources, e.g., (drug, reaction) edges from the drug database, (user, drug) and (user, reaction) edges from the medical record. Here for each dataset that we tested, we used 50% of the edges and only 5% of the hyperedges to train the model. The results are in Fig. 5.

When using only the edges to train the model, our method still achieves higher AUROC and AUPR score for hyperedge prediction as compared to node2vec (Table 2). We found that when the model is trained with both the downsampled hyperedge dataset and the edge dataset, it would be able to reach higher performance or suffer less from overfitting than being trained with each of the datasets individually. This demonstrates that our model can capture the consensus information on the graph structure across different sizes of hyperedges.

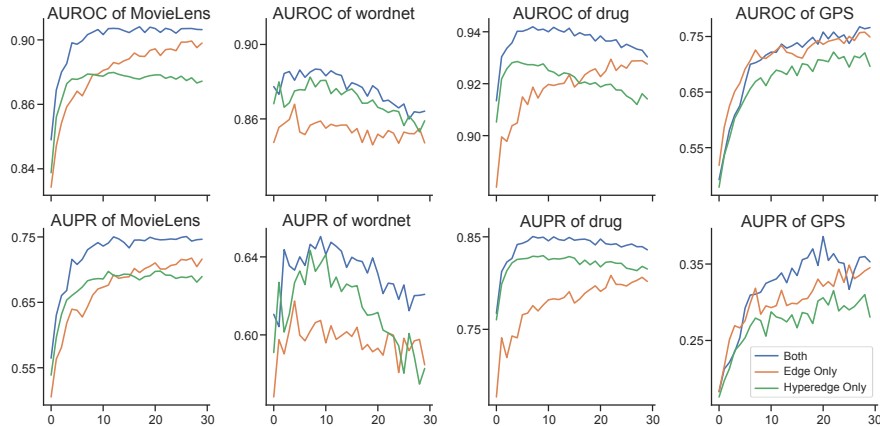

**Figure 5:** AUROC and AUPR scores of Hyper-SAGNN for hyperedge prediction on the downsampled dataset over training epochs.

### 4.3 Outsider Identification

In addition to the standard link prediction and node classification, we further formulated a new task called "outsider identification". Previous methods such as DHNE can answer the question of whether a specific tuple of nodes $(v_1, v_2, ..., v_k)$ form a hyperedge. However, in many settings, we might also want to know the reason why this group of nodes will not form a hyperedge. We first define the outsider of a group of nodes as follows. Node $v_i$ is the outsider of the node group $(v_1, v_2, ..., v_k)$ if it satisfies:

$$\exists e \in E, (v_1, v_2, ..., v_{i-1}, v_{i+1}, ..., v_k) \in e \tag{8}$$

$$\nexists e \in E, s.t. \ \exists j \in \{1, 2, .., k\}, j \neq i, (v_i, v_j) \in e \tag{9}$$

We speculated that Hyper-SAGNN can answer this question by analyzing the probability score $p_1$ to $p_k$ (defined in Eqn. 7). We assume that the node $v_i$ with the smallest $p_i$ would be the outsider. We then set the evaluation as follows. We first trained the model as usual, but at the final stage, we replaced the average pooling layer with the min pooling layer and fine-tuned the model for several epochs. We then fed the generated triplets with known outsider node into the trained model and calculated the top-$k$ accuracy of the outsider node matching the node with the smallest probability. Because this task is based on the prediction results of the hyperedges, we only tested on the dataset that achieves the best hyperedge prediction, i.e., the drug dataset. We found that we have 81.9% accuracy for the smallest probability and 95.3% accuracy for the top-2 smallest probability. These results show that by switching the pooling layer we would have better outsider identification accuracy (from 78.5% to 81.9%) with the cost of slightly decreased hyperedge prediction performance (AUC from 0.955 to 0.935). This demonstrates that our model is able to accurately predict the outsider within the group even without further labeled information. Moreover, the performance of outsider identification can be further improved if we include the cross-entropy between $p_i$ and the label of whether $v_i$ is an outsider for all applicable triplets in the loss term. Together, these results show the advantage of Hyper-SAGNN in terms of the interpretability of hyperedge prediction.

### 4.4 Application to single-cell Hi-C datasets

We next applied Hyper-SAGNN to the recently produced single-cell Hi-C (scHi-C) datasets (Ramani et al., 2017; Nagano et al., 2017). Genome-wide mapping of chromatin interactions by Hi-C (Lieberman-Aiden et al., 2009; Rao et al., 2014) has enabled comprehensive characterization of the 3D genome organization that reveals patterns of chromatin interactions between genomic loci. However, unlike bulk Hi-C data where signals are aggregated from cell populations, scHi-C provides unique information about chromatin interactions at single-cell resolution, thus allowing us to ascertain cell-to-cell variation of the 3D genome organization. Specifically, we propose that scHi-C makes it possible to model the cell-to-cell variation of chromatin interaction as a hyperedge, i.e., (cell, genomic locus, genomic locus). Note that the hyperedege here is "partially non-ordered", namely (cell $i$, locus $j$, locus $k$) should be equivalent to (cell $i$, locus $k$, locus $j$). Our method is able to guarantee that while DHNE cannot achieve that directly. For the analysis of scHi-C, the

most common strategy would be revealing the cell-to-cell variation by embedding the cells based on the contact matrix and then applying the clustering algorithms such as $K$-means clustering on the embedded vectors. We performed the following evaluation to assess the effectiveness of Hyper-SAGNN on learning the embeddings of cells by representing the scHi-C data as hypergraphs.

We tested Hyper-SAGNN on two datasets. The first one consists of scHi-C from four human cell lines: HAP1, GM12878, K562, and HeLa (Ramani et al., 2017). The second one includes the scHi-C that represents the cell cycle of the mouse embryonic stem cells (Nagano et al., 2017). We refer to the first dataset as "Ramani et al. data", and the second as "Nagano et al. data" for abbreviation. We trained Hyper-SAGNN with the corresponding datasets. Due to the large average degrees of cell nodes, the random walk approach would take an extensive amount of time to sample the walks. Thus, we only applied the encoder version of our method. We visualize the learned embeddings by reducing them to 2 dimensions with PCA and UMAP (McInnes et al., 2018) (Fig. 6A-D).

We quantified the effectiveness of the embeddings by applying $K$-means clustering on the Ramani et al. data and evaluating with Adjusted Rand Index (ARI). In addition, we also assessed the effectiveness of the embeddings with a supervised scenario. We used Logistic Regression as the classifier with 10% of the cell as training samples and evaluated the multi-class classification task with Micro-F1 and Macro-F1. We did not run $K$-means clustering on the Nagano et al. data as it represents a state of continuous cell cycle which is not suitable for a clustering task. We instead used the metric ACROC (Average Circular ROC) developed in the HiCRep/MDS paper (Liu et al., 2018) to evaluate the performance of the three methods on the Nagano et al. data. We compared the performance with two recently developed computational methods based on dimensionality reduction of the contact matrix, HiC-Rep/MDS (Liu et al., 2018) and scHiCluster (Zhou et al., 2019). Because Hyper-SAGNN is not a deterministic method for generating embeddings for scHi-C, we repeated the training process 5 times and averaged the score. All these results are in Fig. 6E.

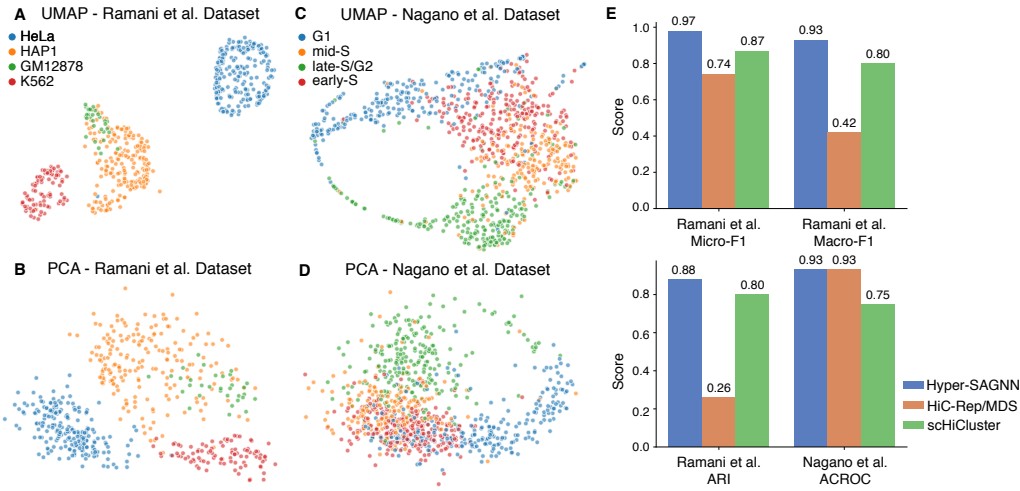

**Figure 6:** **(A)** and **(B):** Visualization of the learned embedding based on Hyper-SAGNN for the Ramani et al. data. **(C)** and **(D):** Visualization of the learned embedding based on Hyper-SAGNN for the Nagano et al. data. Embedding vectors are projected to two dimensional space using either UMAP or PCA. **(E):** Quantitative evaluation of the Hyper-SAGNN on two scHi-C datasets

For the Ramani et al. data (Fig. 6A-B), the visualization of the embedding vectors learned by Hyper-SAGNN exhibits clear patterns that cells with the same cell type are clustered together. Moreover, cell line HAP1, GM12878, and K562 are all blood-related cell lines, which are likely to be more similar to each other in terms of 3D genome organization as compared to HeLa. Indeed, we observed that they are also closer to each other in the embedding space. Quantitative results in Fig. 6E are consistent with the visualization as our method achieves the highest ARI, Micro-F1, Macro-F1 score among all three methods. For the Nagano et al. data, as shown in Fig. 6C-D, we found that the embeddings exhibit a circular pattern that corresponds to the cell cycle. Also, both HiC-Rep/MDS and Hyper-SAGNN achieve high ACROC scores. All these results show the effectiveness of representing the scHi-C datasets as hypergraphs using Hyper-SAGNN, which has great potential to provide insights into the cell-to-cell variation of higher-order genome organization.

## 5 CONCLUSION

In this work, we developed a new graph neural network called Hyper-SAGNN for the representation learning of general hypergraphs. The model has the flexibility to deal with homogeneous and heterogeneous, and uniform and non-uniform hypergraphs. We demonstrated that Hyper-SAGNN can improve or match state-of-the-art performance for hypergraph representation learning while addressing the shortcomings of prior methods such as the inability to predict hyperedges for non-$k$-uniform heterogeneous hypergraphs. Hyper-SAGNN is computationally efficient as the input size to the attention layer is bounded by the maximum hyperedge size as opposed to the number of neighbors.

One potential improvement of Hyper-SAGNN as future work would be to allow information aggregation over all the first-order neighbors before calculating the static/dynamic embeddings for a node with the additional computational cost. With this design, the static embedding for a node would still satisfy our constraint that it is fixed for a known hypergraph with varying input tuples. This would allow us to incorporate previously developed methods on graphs, such as GraphSAGE (Hamilton et al., 2017a) and GCN (Kipf & Welling, 2016), as well as methods designed for hypergraphs like HyperGCN (Yadati et al., 2018) into this framework for better link prediction performance. Such improvement may also extend the application of Hyper-SAGNN to semi-supervised learning.

## ACKNOWLEDGMENT

J.M. acknowledges support from the National Institutes of Health Common Fund 4D Nucleome Program grant U54DK107965, National Institutes of Health grant R01HG007352, and National Science Foundation grant 1717205. Y.Z. (Yao Class, IIIS, Tsinghua University) contributed to this work as a visiting undergraduate student at Carnegie Mellon University during summer 2019.

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

# A  APPENDIX

## A.1  DETAILS OF THE STRATEGIES FOR GENERATING FEATURE VECTORS

We first define the functions used in the subsequent sections as follows: a hyperedge $e$ with weight $w(e)$ is incident with a vertex $v$ if and only if $v \in e$. We denote the indicator function that represents the incident relationship between $v$ and $e$ by $h(v, e)$, which equals 1 when $e$ is incident with $v$. The degree of vertex and the size of hyperedge are defined as:

$$d(v) \triangleq \sum_{e \in E} h(v, e)w(e) \tag{10}$$

$$\delta(e) \triangleq \sum_{v \in V} h(v, e) = |e| \tag{11}$$

### A.1.1  ENCODER BASED APPROACH

As shown on the right side of Fig. 3, the first method to generate features is referred to as the encoder based approach, which is similar to the structure used in DHNE (Tu et al., 2018). We first obtain the incident matrix of the hypergraph $H \in \mathbb{R}^{|V| \times |E|}$ with entries $h(v, e) = 1$ if $v \in e$ and 0 otherwise. We also calculate the diagonal degree matrix $D_v$ containing the vertex degree $d(v) = \sum_{e \in E} h(v, e)$. We thus have the adjacency matrix $A = HH^T - D_v$, of which the entries $a(v_i, v_j)$ denote the concurrent times between each pair of nodes $(v_i, v_j)$. The $i$-th row of $A$, denoted by $\vec{a}_i$, shows the neighborhood structures of the node $v_i$, which then passes through a one-layer neural network to produce $\vec{x}_i$:

$$\vec{x}_i = \tanh\left(W_{\text{enc}} \cdot \vec{a}_i + \vec{b}_{\text{enc}}\right) \tag{12}$$

In DHNE, a symmetric structure was introduced where there are corresponding decoders to transform the $\vec{x}_i$ back to $\vec{a}_i$. Tu et al. (2018) remarked that including this reconstruction error term would help DHNE to learn the graph structure better. We also include the reconstruction error term into the loss function, but with tied-weights between encoder and decoder to reduce the number of parameters that need to be trained.

### A.1.2  RANDOM WALK BASED APPROACH

Besides the encoder based approach, we also utilize a random walk based framework to generate the feature vectors $\vec{x}_i$ (shown on the left side of Fig. 3). We extend the biased 2nd-order random walks proposed in node2vec (Grover & Leskovec, 2016) to generalize to hypergraphs. For a walk from $v$ to $x$ then to $t$, the strategies are described as follows.

The 1st-order random walk strategy given the current vertex $x$ is to randomly select a hyperedge $e$ incident with $x$ based on the weight of $e$ and then to choose the next vertex $y$ from $e$ uniformly (Zhou et al., 2007). Therefore, the 1st-order transition probability is defined as:

$$\pi_1(t|x) \triangleq \sum_{e \in E} w(e) \frac{h(t, e)h(x, e)}{\delta(e)} \tag{13}$$

We then generalize the 2nd-order bias $\alpha_{pq}$ from ordinary graph to hypergraph for a walk from $v$ to $x$ to $t$ as:

$$\alpha_{p,q}(t, v) = \begin{cases} 1/p, & \text{if } \exists e \in E, \text{ s.t. } t, v, x \in e \\ 1, & \text{else if } \exists e \in E, \text{ s.t. } t, x \in e \\ 1/q, & \text{otherwise} \end{cases} \tag{14}$$

where the parameters $p$ and $q$ are to control the tendencies that encourage outward exploration and obtain a local view.

Next we add the above terms to set the biased 2nd-order transition probability as:

$$\pi(t|v, x) = \begin{cases} \frac{\pi_1(t|x) \cdot \alpha_{pq}(t,v)}{Z}, & \text{if } \exists e \in E, \text{ s.t. } v, x \in e \\ 0, & \text{otherwise} \end{cases} \tag{15}$$

where $Z$ is a normalizing factor.

With the well-defined 2nd-order transition probability $\pi(t|v,x)$, we simulate a random walk of fixed length $l$ through a 2nd-order Markov process marked by $P(c_i = t|c_{i-1} = x, c_{i-2} = v) = \pi(t|v,x)$, where $c_i$ is the $i$-th node in the walk. A Skip-gram model (Mikolov et al., 2013; Mikolov et al., 2013) is then used to extract the node features from sampled walks such that the nodes that appear in similar contexts would have similar embeddings.

## A.2 DETAILS OF THE DATASETS USED IN THIS WORK

The four datasets used in the first part of our evaluation are:

- GPS (Zheng et al., 2010): GPS network. The hyperedges are based on (user, location, activity) relations.
- MovieLens (Harper & Konstan, 2015): Social network. The hyperedges are based on (user, movie, tag) relations, describing peoples' tagging activities.
- drug: Medicine network from FAERS[1]. The hyperedges are based on (user, drug, reaction) relations.
- wordnet (Bordes et al., 2013): Semantic network from WordNet 3.0. The hyperedges are based on (head entity, relation, tail entity), expressing the relationships between words.

Details of the datasets are shown in Table A1.

**Table A1:** Datasets used in this work. Note that the columns under "#(V)" correspond to the columns under "node type" for each dataset.

| DATASETS | NODE TYPE | | | #(V) | | | #(E) |
|---|---|---|---|---|---|---|---|
| GPS | user | location | activity | 146 | 70 | 5 | 1,436 |
| MovieLens | user | movie | tag | 2,113 | 5,908 | 9,079 | 47,957 |
| drug | user | drug | reaction | 12 | 1,076 | 6,398 | 171,756 |
| wordnet | head | relation | tail | 40,504 | 18 | 40,551 | 145,966 |

## A.3 PARAMETER SETTING

We downloaded the source code of DHNE from its GitHub repository. The structure of the neural network of DHNE was set to be the same as what the authors described in Tu et al. (2018). We tuned parameters such as the $\alpha$ term and the learning rate following the same procedure. We also tried adding dropout between representation vectors and the fully connected layer for better performance of DHNE. All these parameters were tuned until it was able to replicate or even improve the performance reported in the original paper. To make a fair comparison, for all the results below, we made sure that the training and validation data setups were the same across different methods.

For node2vec, we decomposed the hypergraph into pairwise edges and ran node2vec on the decomposed graph. For the hyperedge prediction task, we first used the learned embedding to predict pairwise edges. We then used the mean or min of the pairwise similarity as the probability for the tuple to form a hyperedge. We set the window size to 10, walk length to 40, the number of walks per vertex to 10, which are the same parameters used in DHNE for node2vec. However, we found that for the baseline method node2vec, when we tuned the hyper-parameter $p, q$ and also used larger walk length, window size and walks per vertex (120, 20, 80 instead of 40, 10, 10), it would achieve comparable performance for node classification task as DHNE. This observation is consistent with our designed biased hypergraph random walk. But this would result in a longer time for sampling the walks and training the skip-gram model. We therefore kept the parameters consistent with what was used in DHNE paper.

For our Hyper-SAGNN, we set the representation size to 64, which is the same as DHNE. The number of heads in the multi-head attention layer is set to 8. When using the encoder based approach to calculate $x_i$, we set the encoder structure to be the same as the encoder part in DHNE. When using the random walk based approach, we decomposed the hypergraph into a graph as described above. We set the window size to 10, walk length to 40, the number of walks per vertex to 10, to allow time-efficient generation of feature vector $\vec{x}_i$. The results in Section 4.1 showed that even when the

---

[1]http://www.fda.gov/Drugs/

pre-trained embeddings are not so ideal, Hyper-SAGNN can still well capture the structure of the graph.

To train the model, we used the Adam optimizer with learning rate 1e-3. Each batch contains 96 positive hyperedges with 480 negative samples. The training is terminated when it reaches the maximum training epoch number (100) or the performance on the validation set no longer improves.

## A.4 EXPERIMENT SETTING

To compare the performance of Hyper-SAGNN with other baseline methods, we used the same three tasks in the DHNE paper, i.e., network reconstruction, link prediction, and node classification. In this section, we describe the setting of these three tasks and the evaluation metrics.

Network reconstruction aims to reconstruct the input hypergraph from the learned embedding. Specifically, a hypergraph $G = (V, E)$ is used as input to the algorithm. After training, the model makes predictions for the original hyperedge set $E$. Link prediction, on the other hand, aims to predict the unseen hyperedge set $E^{'}$ based on the model trained with hypergraph $G = (V, E)$. These two tasks can be regarded as binary classification tasks and thus can be evaluated by metrics such as AUROC and AUPR. For both tasks, the negative samples are set to be 5 times the amount of the positive samples following the same setup of DHNE.

For the node classification, after training the model, the embeddings for the nodes are used to train a Logistic Regression classifier with targets as the pre-defined labels for the nodes. Here we have a multi-label classification and a multi-class classification task for dataset MovieLens and wordnet, respectively, making the metrics defined for binary classification not applicable. Therefore, the performances are evaluated by Micro and Macro F1 scores that are used for quantifying the performance of multi-label/multi-class classification.

## A.5 ANALYSIS OF THE DYNAMIC EMBEDDINGS

In this section, we discuss the relationships between dynamic embeddings and static embeddings. As mentioned in the Method section, we design the model to establish the connection between the probability score for a tuple with how well the static embedding of each node can be approximated by the features of their neighbor within that tuple. If the model works as we designed, the dynamic embedding of node $i$ conditioned on an actual hyperedge $(i, j, ..., k)$ should approximate the static embedding of node $i$ better. In contrast, the dynamic embedding of node $i$ when it is a member of a non-hyperedge tuple would not have a good "approximation". To evaluate this, after training the model we collected all the dynamic embeddings for nodes when they are within the tuples of positive samples, which would be referred to as positive dynamic embeddings. We also collected the dynamic embeddings for nodes when conditioned on the "hard negative samples" (the negative samples that are generated by only changing one node in the positive samples), which would be referred to as negative dynamic embeddings. We tested whether the positive dynamic embeddings indeed resemble the static embeddings better by comparing the performance of node classification using dynamic embeddings and static embeddings. For each node, since the number of positive and negative dynamic embeddings is not finite, we sampled and averaged a number of them as the features. The remaining setting of node classification is the same as described in the main text. As shown in Fig. A1, in general, the positive dynamic embeddings can achieve much better performance as compared to the negative dynamic embeddings. Both the micro and macro F1 scores increase when more positive dynamic embeddings are sampled and averaged. When averaging all positive dynamic embeddings and using those as features, the node classification performance is close to what we achieved using static embeddings. This analysis demonstrates that the positive dynamic embeddings "approximate" the static embeddings as they contain sufficient information for accurate node classification. In addition, for the negative dynamic embeddings where only one node is changed when generating negative samples, it performs dramatically worse. Note that we exclude the $\alpha_{ii}$ term in the calculation of dynamic embeddings, which makes the dynamic embedding for node $i$ the combination of features of its neighbor within a given tuple. These results demonstrate that the probability scores for each node indicate the "distances" of static/dynamic embedding pairs that reflect how well the static embedding of each node can be approximated by the features of their neighbor within that tuple.

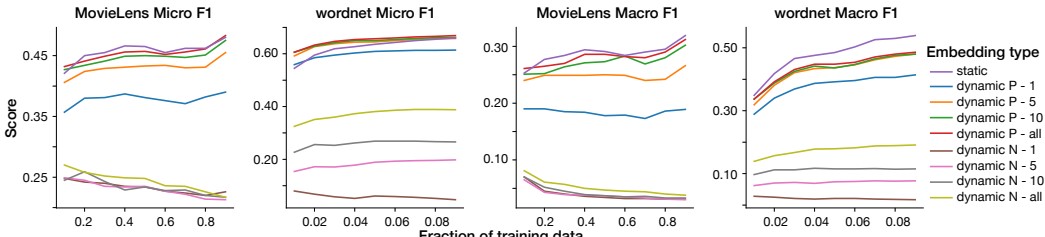

**Figure A1:** Node classification performance comparison for static embeddings and dynamic embeddings. The results based on static embeddings are marked as "static". Results based on positive/negative dynamic embeddings are marked as "dynamic P/dynamic N". The numbers of dynamic embeddings sampled during this process are also included in the legend.

## A.6  COMPARISON OF HYPER-SAGNN VS. THE VARIANTS

As mentioned above, unlike the standard GAT model, we exclude the $\alpha_{ii}$ term in the self-attention mechanism. To test whether this constraint would improve or reduce the model's ability to learn, we implemented a variant of our model (referred to as variant type I) by including this term. Also, as mentioned in the Method section, another potential variant of our model would be directly using the $\vec{d}_i^*$ to calculate the probability score $p_i^*$. We refer to this variant as variant type II. For variant type II, on node classification task, since it does not have a static embedding, we used $W_v^T x_i$. The rest of the parameters and structure of the neural network remain the same.

We then compared the performance of Hyper-SAGNN and two variants in terms of AUC and AUPR values for network reconstruction task and hyperedge link prediction task on the following four datasets: MovieLens, wordnet, drug, and GPS. We also compared the performance in terms of the Micro F1 score and Macro F1 score on the node classification task on the MovieLens and the wordnet dataset. For the MovieLens dataset, we used 90% nodes as training data while for wordnet, we used 1% of the nodes as training data. All the evaluation setup is the same as described in the main text. To avoid the effect of randomness from the neural network training, we repeated the training process for each experiment five times and made the line plot of the score versus the epoch number. To illustrate the differences more clearly, we started the plot at epoch 3 for the random walk based approach and epoch 12 for the encoder based approach. The performance of the model using the random walk based approach is shown in Fig. A2 to Fig. A5. The performance of the model using the encoder based approach is shown in Fig. A6 to Fig. A9.

For models with the random walk based approach, Hyper-SAGNN is the best in terms of all metrics for the GPS, MovieLens, and wordnet dataset. On the drug dataset, Hyper-SAGNN achieves higher AUROC and AUPR score on the network reconstruction task than two variants, but slightly lower AUROC score for the link prediction task (less than 0.5%).

For models with the encoder based approach, the advantage is not that obvious. All 3 methods achieve similar performance in terms of all metrics for the GPS and the drug dataset. For the Movie-Lens and wordnet dataset, Hyper-SAGNN performs similar to variant type I, higher than variant type II on the network reconstruction and link prediction task. However, our model achieves slightly higher accuracy on the node classification task than variant type I.

Therefore, these evaluations show that the choice of the structure of Hyper-SAGNN can achieve higher or at least comparable performance than the two potential variants over multiple tasks on multiple datasets.

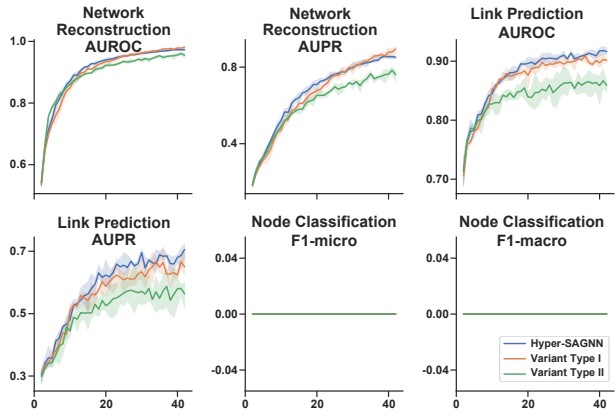

**Figure A2:** Performance comparison of Hyper-SAGNN – Walk and Variant Type I, II (GPS)

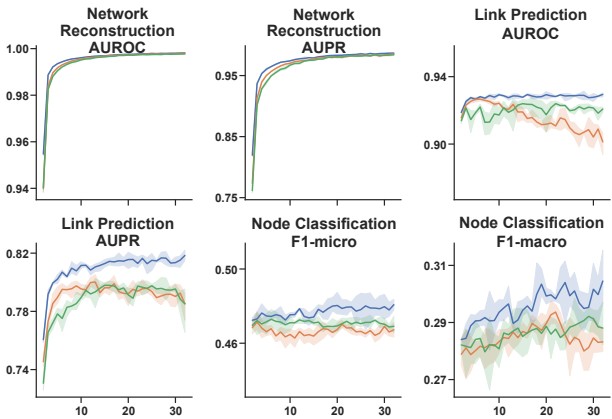

**Figure A3:** Performance comparison of Hyper-SAGNN – Walk and Variant Type I, II (MovieLens)

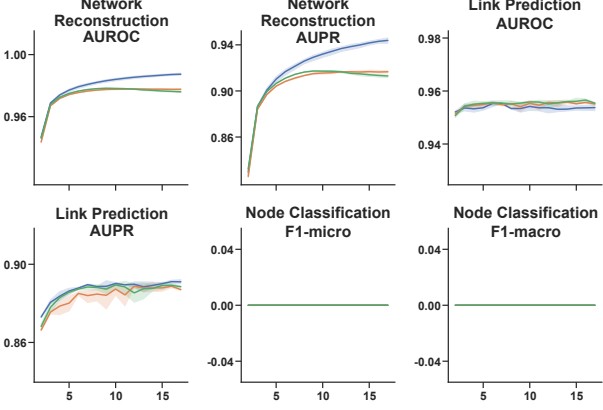

**Figure A4:** Performance comparison of Hyper-SAGNN – Walk and Variant Type I, II (drug)

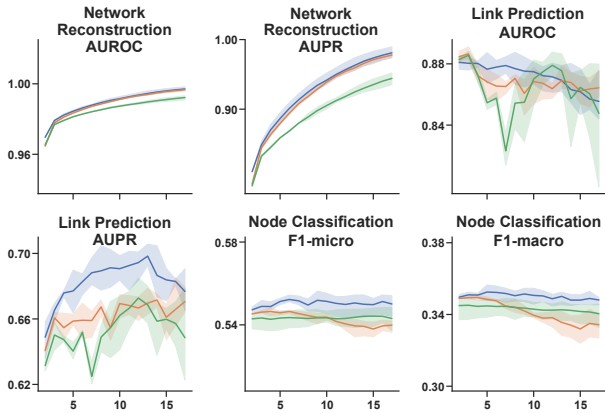

**Figure A5:** Performance comparison of Hyper-SAGNN – Walk and Variant Type I, II (wordnet)

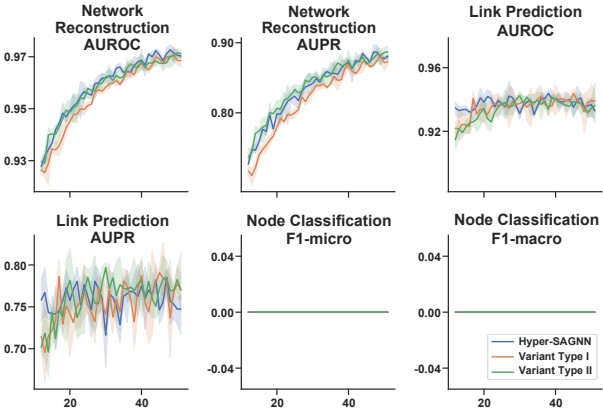

**Figure A6:** Performance comparison of Hyper-SAGNN – Encoder and Variant Type I, II (GPS)

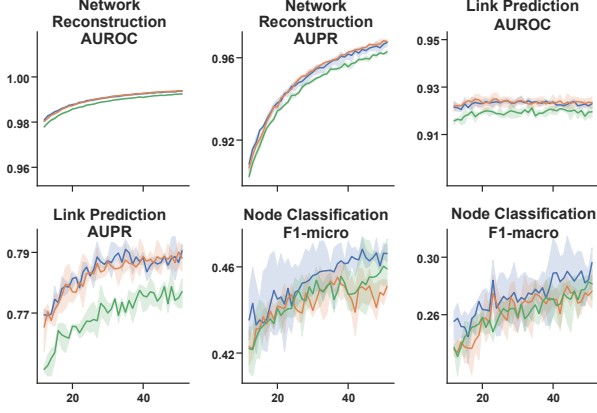

**Figure A7:** Performance comparison of Hyper-SAGNN – Encoder and Variant Type I, II (MovieLens)

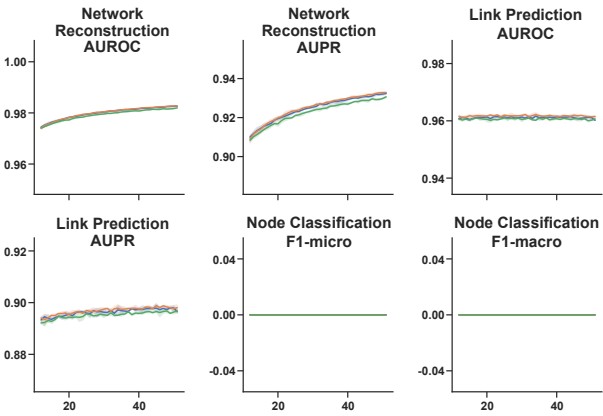

**Figure A8:** Performance comparison of Hyper-SAGNN – Encoder and Variant Type I, II (drug)

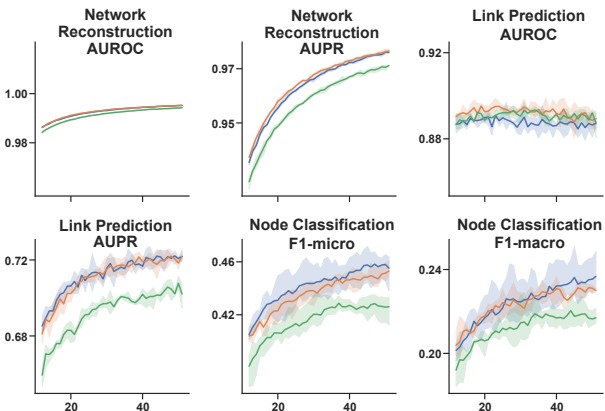

**Figure A9:** Performance comparison of Hyper-SAGNN – Encoder and Variant Type I, II (wordnet)

