# OpenReview forum: "Hyper-SAGNN: a self-attention based graph neural network for hypergraphs"
_ICLR.cc/2020/Conference — Accept (Poster)_

### Official Review · AnonReviewer3 · 2019-10-23
**Official Blind Review #3**

**Rating:** 8

**Review:**

Summary:

This work is tackling the problem of predicting variable-sized heterogeneous hyperedges in hypergraphs. The authors do so by introducing a multi-head attention mechanism that takes in a set of node features and predicts the probability of this set of nodes forming a hyperedge.

- Quality and clarity:

The quality of the paper is good, the writing is clear and the reasoning is relatively easy to follow. The problem the authors are trying to tackle is also clearly defined. The motivation of the dynamic and static embedding choice could be explained better. The intuition that the pseudo-distance tells us how well the static embedding
of a node can be approximated by the features of the neighbour within the tuple somewhat makes sense, but I also have the impression that this is just one possible interpretation and what happens in practice could actually be quite different. A more formal treatment and analysis of this intuition would be very helpful.

- Originality:
The main contribution here is to propose a more flexible formulation to improve over the fixed MLP used by DHNE. This in itself is a good idea, but overall is an incremental improvement.

- Significance:
The results are convincing, DHNE is clearly outperformed on the benchmarks. Depending on the ease of training the model compared to DHNE, this work might be a useful contribution for the ICLR community.

Other points

The related work section should be expanded by discussing relationships to:

Bai, Song, Feihu Zhang, and Philip HS Torr. "Hypergraph Convolution and Hypergraph Attention." arXiv preprint arXiv:1901.08150 (2019).

Feng, Yifan, et al. "Hypergraph neural networks." Proceedings of the AAAI Conference on Artificial Intelligence. Vol. 33. 2019.

------------------
Post-rebuttal
------------------

I would like to thank the authors for their in-depth response. I found the additional analysis on the dynamic node embeddings insightful and it is reassuring that it confirms the claims. Overall I think the rebuttal addresses my major concerns and I will raise my score.

**Experience Assessment:**

I do not know much about this area.

**Review Assessment: Checking Correctness Of Derivations And Theory:**

I assessed the sensibility of the derivations and theory.

**Review Assessment: Checking Correctness Of Experiments:**

I assessed the sensibility of the experiments.

**Review Assessment: Thoroughness In Paper Reading:**

I read the paper thoroughly.

---

> ### Author Response · Authors · 2019-11-15
> **Response to Reviewer 3**
>
>
> Response: We would like to thank the reviewer for all the great suggestions. We have thoroughly addressed the questions and provided point-by-point responses to the reviewer’s comments.
>
> >>> The quality of the paper is good, the writing is clear and the reasoning is relatively easy to follow. The problem the authors are trying to tackle is also clearly defined. The motivation of the dynamic and static embedding choice could be explained better. The intuition that the pseudo-distance tells us how well the static embedding of a node can be approximated by the features of the neighbour within the tuple somewhat makes sense, but I also have the impression that this is just one possible interpretation and what happens in practice could actually be quite different. A more formal treatment and analysis of this intuition would be very helpful.
>
> Response: We appreciate this question from the reviewer. To address the reviewer's concern, we did additional analysis on our original rationale that the pseudo-distance tells us how well the static embedding of a node can be approximated by the features of the neighbor within the tuple. We added a section called "Analysis of the dynamic embeddings" (A.5) in the Appendix. Briefly, we collected the dynamic embeddings for each node when it is presented in the positive samples and negative samples, respectively. We referred to them as positive dynamic embeddings and negative dynamic embeddings, respectively. We found that when using the sampled and averaged dynamic embeddings for the node classification task, the positive ones can achieve similar performance as compared to the static embeddings while the negative ones perform much worse. This demonstrates that the positive dynamic embeddings "approximate" the static embeddings better as compared to the negative ones. These additional analyses strengthened our manuscript in terms of further clarifying the motivation of the static/dynamic embeddings design.
>
> >>> Originality: The main contribution here is to propose a more flexible formulation to improve over the fixed MLP used by DHNE. This in itself is a good idea, but overall is an incremental improvement.
>
> Response: We thank the reviewer for this comment. We would like to clarify that, to generalize the hyperedge prediction from fixed type hyperedges to general cases, we proposed a self-attention based neural network to replace the fixed MLP that also improves the performances on all tasks that the previous methods evaluated. Nevertheless, our Hyper-SAGNN is not limited to simply improve the performances of the tasks evaluated in DHNE. We believe that Hyper-SAGNN can be used for new tasks and applications such as the outsider identification that we evaluated in Section 4.3. Additionally, the dynamic/static embeddings design also offers a new perspective for analyzing the properties of instances within a hypergraph. Taken together, we believe that these new applications and analyses demonstrate the novelty of our work.
>
> >>>  Significance: The results are convincing, DHNE is clearly outperformed on the benchmarks. Depending on the ease of training the model compared to DHNE, this work might be a useful contribution for the ICLR community.
>
> Response: We would like to thank the reviewer for the positive comments. As for the potential concern about the ease of training the model, we would like to point out that the model is robust and easy to train. We now reported the training hyperparameters in the Appendix to further clarify. We used the Adam optimizer with default settings except for a smaller learning rate (1e-3). We found that the performance is robust to the choice of hyperparameters such as the number of heads in the multi-head attention layer.
>
> >>> Other points
>
> The related work section should be expanded by discussing relationships to:
>
> Bai, Song, Feihu Zhang, and Philip HS Torr. "Hypergraph Convolution and Hypergraph Attention." arXiv preprint arXiv:1901.08150 (2019).
>
> Feng, Yifan, et al. "Hypergraph neural networks." Proceedings of the AAAI Conference on Artificial Intelligence. Vol. 33. 2019.
>
> Response: We thank the reviewer for pointing out these prior work. We added references of these two papers and discussed them in the related work section. These methods generalize the convolution and attention operation of graphs to hypergraphs and are typically used for semi-supervised node classification task on hypergraphs with known node attributes. In contrast, our new method Hyper-SAGNN is designed to learn the function for hyperedge prediction and can be used for hypergraphs without node attributes.

---

### Official Review · AnonReviewer2 · 2019-10-24
**Official Blind Review #2**

**Rating:** 8

**Review:**

This paper proposes a new graph neural network model capable of performing tasks over hyperedges of variable type and size (i.e. different number and types of graph nodes connected by different hyperedges).  They experimentally verify its effectiveness over the previous state of the art on several datasets and tasks.

I lean toward accepting this paper.  It is a relevant topic area, the design appears novel relative to recent work, and the presentation is mostly clear.  The design decisions are well-motivated and discussed (e.g. the choices to use the static embedding, the exclusion of the diagonal self-attention weights).  Performance is tested on previously used datasets and tasks, for a thorough comparison against recent best methods (DHNE).  The model is clearly described.

A possible weakness of this paper is that the evaluation data sets are all k-uniform hypergraphs with k=3.  This is perhaps the minimal case which their method can address.  For all the discussion of generality to heterogeneous hyperedges, it would have been better to include some dataset (even if synthetic) to establish baseline performance over while ablating k>3 and multiple hyperedge types.  This remains completely un-investigated (even the genome dataset appears to be k=3?).  Although, their edge/hyperedge ablation on these datasets partially addresses the point and the result (Fig 5) seems promising.

Some other notes:
Eq 5.  Given that it is the default, explicitly write that the summation is over j!=i
Eq 8.  Probably leave the ellipses out between v_{i-1} and v_{i+1}?
The tasks and measures like AUC, AUPR, and network reconstruction could be described somewhere, even if in an appendix.
Some/many training hyperparameters not listed?


EDIT: Post rebuttal, authors have provided evidence of good performance on data sets with k=4 and k=5, addressing my main concern, and have addressed the smaller notes.  Raising score but still am not expert in this area.

**Experience Assessment:**

I do not know much about this area.

**Review Assessment: Checking Correctness Of Derivations And Theory:**

I assessed the sensibility of the derivations and theory.

**Review Assessment: Checking Correctness Of Experiments:**

I assessed the sensibility of the experiments.

**Review Assessment: Thoroughness In Paper Reading:**

I read the paper at least twice and used my best judgement in assessing the paper.

---

> ### Author Response · Authors · 2019-11-15
> **Response to Reviewer 2**
>
> Response: We would like to thank the reviewer for the important suggestions. We have thoroughly addressed the questions and provided point-by-point responses to the reviewer’s comments.
>
> >>> A possible weakness of this paper is that the evaluation data sets are all k-uniform hypergraphs with k=3.  This is perhaps the minimal case which their method can address.  For all the discussion of generality to heterogeneous hyperedges, it would have been better to include some dataset (even if synthetic) to establish baseline performance over while ablating k>3 and multiple hyperedge types.  This remains completely un-investigated (even the genome dataset appears to be k=3?).  Although, their edge/hyperedge ablation on these datasets partially addresses the point and the result (Fig 5) seems promising.
>
> Response: We thank the reviewer for raising this point. Hyper-SAGNN can handle hyperedges with size larger than 3 because both the multi-head attention layer used for generating the dynamic embeddings and the position-wise feedforward neural network used for generating static embeddings can take variable sized input. The design in our model for aggregating static/dynamic embeddings are not specific to the size of hyperedges as well. As a proof-of-principle, we demonstrated the generality of Hyper-SAGNN for different types of hyperedges as compared to the 4 datasets used in DHNE.
>         The first experiment, as the reviewer mentioned, is providing the model with a few hyperedges and abundant edges. As shown in Fig. 5, the model can make accurate predictions on both the edges and hyperedges showing that the model can learn from hyperedges with different sizes. The second evaluation is the genomic dataset (i.e., single-cell Hi-C data) we used in this work. Although being a 3-uniform hypergraph, it has a unique property as compared to the other datasets in this work. The hyperedge defined in this dataset is "(cell, genomic locus, genomic locus)",  which is partially non-ordered. In other words, the hyperedge (cell i, locus j , locus k) is equivalent to the hyperedge (cell i, locus k , locus j). Our model can guarantee this property while DHNE cannot achieve it directly. We now have added this to our manuscript to clarify this particular point in the evaluation.
>
> >>> Some other notes:
>        Eq 5.  Given that it is the default, explicitly write that the summation is over j!=i
>        Eq 8.  Probably leave the ellipses out between v_{i-1} and v_{i+1}?
>        The tasks and measures like AUC, AUPR, and network reconstruction could be described somewhere, even if in an appendix.
>        Some/many training hyperparameters not listed?
>
> Response: We apologize for the confusion. We have changed the equations following the reviewer's suggestion. We added a section called "Experiment Setting" in the Appendix to better describe the evaluation tasks and metrics. We now also listed the training hyperparameters including the optimizer, learning rate, batch size, etc in the "Parameter setting" section in the Appendix.

---

### Author Response · Authors · 2019-11-15
**Summary of revisions made to the paper**

We would like to thank the reviewers for all the comments and suggestions. We have made revisions to the paper to address all the constructive comments. Our major changes include:
- In section 2, we now mentioned more related works as suggested by the reviewer and highlighted the differences and novelty in our work.
- In section 4.4, we have added a sentence to clarify the difference between the hyperedges of the scHi-C datasets with the other hypergraph datasets used in this work.
- In the Appendix section A.3, we have added more hyperparameters related to the training of our method.
- We have added a section in the Appendix (A.4) to describe the evaluation tasks and the metrics used in this work.
- We have added a section in the Appendix (A.5) to describe the analysis of the dynamic embeddings to explain our rationale of the static/dynamic embedding design more clearly.

---

### Public Comment · ~Ted_Liu1 · 2019-12-20
**Some recent work on hypergraph embedding not mentioned in the paper**

Heterogeneous Hyper-Network Embedding, ICDM'18
Modeling Multi-way Relations with Hypergraph Embedding, CIKM'18
Hyper-Path-Based Representation Learning for Hyper-Networks, CIKM'19

---

### Author Response · Authors · 2020-04-16
**Code for Hyper-SAGNN**

The code for Hyper-SAGNN is now maintained under: https://github.com/ma-compbio/Hyper-SAGNN

---

### Decision · Program_Chairs · 2019-12-19

**Decision:**

Accept (Poster)

**Comment:**

This work introduces a new neural network model that can represent hyperedges of variable size, which is experimentally shown to improve or match the state of the art on several problems.

Both reviewers were in favor of acceptance given the method's strong performance, and had their concerns resolved by the rebuttals and the discussion. I am therefore recommending acceptance.